# KnoBuilder: An LLM-Agent for Autonomous and Personalized Knowledge Graph Construction from Unstructured Text

## Abstract

This paper introduces KnoBuilder, a novel LLM-based agentic framework for autonomous construction of personalized knowledge graphs from unstructured text corpora. Addressing the limitations of traditional knowledge graph construction methods and one-shot LLM extraction approaches, KnoBuilder implements a synergistic loop between an LLM agent and a dynamically evolving knowledge graph. The framework features strategic planning for knowledge acquisition, self-refining information extraction with multi-stage validation, and dynamic consolidation maintaining graph coherence. Extensive evaluation on scientific corpora demonstrates that KnoBuilder significantly outperforms state-of-the-art baselines, achieving 85% F1-score in extraction quality, 46% improvement in acquisition efficiency, 91% entity resolution accuracy, and superior performance in complex query answering, while maintaining coherent graph structures with 96% consistency scores.

## 1 Introduction

The proliferation of digital text has created an unprecedented opportunity to compile human knowledge into structured, machine-readable formats. Knowledge Graphs (KGs)—structured representations of entities and their relationships—have emerged as a powerful paradigm for achieving this, enabling complex reasoning, semantic search, and data integration in applications from web search to enterprise analytics Singhal [2012], Ji et al. [2021]. However, the construction and curation of KGs have traditionally relied on manual effort, crowdsourcing, or brittle, domain-specific extraction pipelines, creating a significant bottleneck that limits their scalability, dynamism, and personalization Paulheim [2017]. This "KG bottleneck" is particularly acute in domains like scientific research, where the volume of new literature rapidly outpaces the capacity of human curators, making it difficult for individuals to maintain a personalized, connected view of their field.

Large Language Models (LLMs) offers a potential solution to this problem. With their profound ability to understand and generate natural language, LLMs can, in principle, identify entities and relations within text to populate a KG Yao et al. [2023a]. Yet, a critical gap exists between this capability and the reliable, autonomous construction of a high-quality, coherent KG. A naive application of LLMs for one-shot information extraction often results in outputs plagued by inconsistency, hallucination, and a lack of strategic focus Ji et al. [2023]. An LLM alone does not embody the sustained, goal-directed process of a human knowledge curator, who strategically seeks out information, cross-references new facts with existing knowledge, and maintains a consistent, integrated mental model. This gap necessitates a shift from treating the LLM as a simple extractor to positioning it as the core of an **agentic system**—an autonomous AI that can plan, execute, and refine a sequence of actions to achieve a complex goal Xi et al. [2023].

Submitted to 39th Conference on Neural Information Processing Systems (NeurIPS 2025). Do not distribute.

In this paper, we introduce **KnoBuilder**, a framework that operationalizes this vision by using an LLM-powered agent to autonomously construct a personalized Knowledge Graph from a large corpus of unstructured text. The core of our approach lies in a synergistic loop between the agent and the KG it is building. The agent, guided by a **user profile**, performs key cognitive functions: it *plans* a knowledge acquisition strategy by generating relevant search queries, *filters* incoming documents for relevance, *extracts* structured triples using a defined schema, *self-corrects* potential errors by validating extractions against the source text, and *consolidates* new knowledge by linking it to existing entities within the growing graph. The KG, in turn, serves as the agent's structured, external memory, accumulating semantic knowledge and enabling complex, multi-hop reasoning that would be impossible from the raw text alone.

## 2 Literature Review

Traditional approaches to KG construction have often relied on supervised learning for Named Entity Recognition (NER) and Relation Extraction (RE). Pioneering methods like the CNN-based model of Zeng et al. [2014] and the graph LSTM approach of Guo et al. [2019] demonstrated the feasibility of automatically extracting relational facts. To scale this process, open information extraction (OpenIE) paradigms were developed to extract relations without a pre-defined schema Etzioni et al. [2008]. However, these supervised methods are inherently limited by their reliance on large, annotated training datasets, which are expensive to produce and often domain-specific, making them brittle when faced with new entity types or relational phrases Ji et al. [2021]. LLMs, with their extensive world knowledge and powerful in-context learning capabilities, can perform NER and RE in a zero-shot or few-shot manner, significantly reducing the need for task-specific training data Yao et al. [2023a]. Prompting strategies have been developed to guide LLMs for structured knowledge extraction, such as the work by Wei et al. [2022] on chain-of-thought prompting, which improves reasoning for complex extraction tasks. Furthermore, frameworks like UIE Lu et al. [2022] and REBEL Huguet Cabot and Navigli [2021] have sought to unify information extraction tasks, demonstrating the potential of generative models for populating knowledge bases. The capability of LLMs to generate knowledge directly has also been explored, as seen in the knowledge probing studies of Petroni et al. [2019] and the subsequent creation of generated KGs Sen et al. [2024].

The concept of an LLM acting as an autonomous agent that uses tools and plans over a sequence of actions was popularized by frameworks like ReAct Yao et al. [2022], where interleaving reasoning and action steps leads to more robust task performance. This has been extended in more sophisticated agent architectures like AutoGPT Richard and Grolig [2023], LangChain Chase [2022], and LangGraph, which enable complex, stateful workflows. A key application of such agents has been in augmenting their capabilities through external knowledge, most prominently through Retrieval-Augmented Generation (RAG) Lewis et al. [2020]. While RAG typically retrieves unstructured text snippets, recent work on Graph RAG Microsoft [2024] has begun to explore using sub-graphs from a knowledge base to provide more structured and connected context, enhancing the reasoning capabilities of LLMs.

The interplay between agents and knowledge graphs is a nascent but rapidly evolving area. Yao et al. [2023b] demonstrated how an LLM agent could decompose complex questions and leverage a KG for multi-hop reasoning. Similarly, Pan et al. [2023] explored using KGs to guide the planning of LLM agents. On the construction side, Tan et al. [2024] and Lee et al. [2024] have proposed methods where LLM agents iteratively build and refine knowledge graphs, acknowledging the iterative nature of the task. The broader vision of using LLMs for data management, including KG construction, has been articulated in surveys like that of Fan et al. [2023], Li [2025]. However, these emerging approaches often treat the KG as a static resource or focus on a single aspect of the problem, such as extraction or reasoning, in isolation.

Despite this progress, a significant gap remains in the literature. Most existing methods do not fully integrate the *entire pipeline* of personalized KG construction—from strategic knowledge acquisition and relevance filtering to self-corrected extraction and dynamic consolidation—within a single, autonomous agentic loop. Approaches that rely on one-shot LLM extraction lack the iterative refinement and strategic planning necessary for building a coherent, large-scale graph Ji et al. [2023]. Meanwhile, agentic systems that use KGs often assume the graph is pre-existing and static, rather than being collaboratively built and refined by the agent itself during its operation. KnoBuilder aims to address this gap by proposing a holistic, agent-centric framework where the LLM agent and the KG co-evolve.

# 3 Methodology

## 3.1 Mathematical Framework and Problem Formulation

We formalize the personalized KG construction task as a sequential optimization problem where an agent interacts with a document corpus $\mathcal{D}$ to build a target knowledge graph $\mathcal{G}^*$ that maximizes alignment with a user profile $\mathcal{P}$. Let $\mathcal{G}_t = (V_t, E_t, \phi_t)$ represent the KG at time step $t$, where $V_t$ is the set of entities, $E_t \subseteq V_t \times R \times V_t$ is the set of typed relations from relation set $R$, and $\phi_t : V_t \cup E_t \to \mathbb{R}^d$ denotes the feature mapping. The user profile $\mathcal{P}$ is modeled as a multivariate distribution over topics, entities, and relations of interest, which can be explicitly specified or learned from interaction history. The objective function maximizes the expected utility of the constructed graph over a planning horizon $T$:

$$\max_{\pi} \mathbb{E}\left[\sum_{t=0}^{T} \gamma^t U(\mathcal{G}_t, \mathcal{P}) - \lambda C(\pi_t)\right] \tag{1}$$

where $\pi$ represents the agent's policy, $\gamma \in [0, 1]$ is a discount factor, $U(\cdot)$ measures the utility of the graph given the profile, $C(\cdot)$ represents the computational cost of action $\pi_t$, and $\lambda$ controls the cost-utility trade-off. The utility function decomposes into coverage, coherence, and personalization components:

$$U(\mathcal{G}, \mathcal{P}) = \alpha \cdot \text{Coverage}(\mathcal{G}, \mathcal{P}) + \beta \cdot \text{Coherence}(\mathcal{G}) + \eta \cdot \text{Personalization}(\mathcal{G}, \mathcal{P}) \tag{2}$$

with $\alpha + \beta + \eta = 1$ as weighting parameters. The coverage metric measures how well $\mathcal{G}$ captures domain knowledge relevant to $\mathcal{P}$, coherence quantifies the structural consistency and connectivity of the graph, and personalization assesses alignment with user-specific interests. This formulation advances beyond existing work by explicitly modeling the trade-offs between exploration of new knowledge and exploitation of existing structures, addressing the strategic deficiency in one-shot extraction methods Ji et al. [2023]. The optimization proceeds through iterative application of our agentic workflow, which we formalize as a partially observable Markov decision process where the agent maintains a belief state over the true knowledge distribution and selects actions to reduce uncertainty while maximizing utility.

## 3.2 Agentic Workflow Architecture

The KnoBuilder architecture implements the mathematical framework through a structured workflow comprising four interconnected modules that operate in a cyclic manner, as illustrated in Figure 1. Unlike traditional pipeline approaches that process documents in isolation Lu et al. [2022], our system maintains state across iterations and uses the growing KG to inform subsequent decisions, creating a virtuous cycle of knowledge refinement. The Planning Module generates strategic queries based on both the user profile and the current state of the graph, identifying knowledge gaps and promising exploration directions. The Filtering Module then processes retrieved documents, assessing relevance using a multi-faceted scoring function that considers topical alignment, novelty relative to existing knowledge, and source credibility. Relevant documents proceed to the Extraction Module, where our novel self-correcting mechanism produces structured triples while minimizing hallucinations and inconsistencies. Finally, the Consolidation Module integrates new knowledge into the existing graph, resolving entity conflicts and reinforcing connections through semantic similarity measures. This architecture directly addresses the coordination deficiency in existing agentic systems Xi et al. [2023] by maintaining a unified state representation that guides all modules toward the collective objective of building a high-quality, personalized KG. The workflow continues until either a predefined resource budget is exhausted or the graph quality metrics stabilize, indicating sufficient coverage of the target knowledge domain.

## 3.3 Strategic Planning and Knowledge Acquisition

The planning module addresses the critical limitation of passive document processing in traditional IE systems by proactively guiding knowledge acquisition toward high-value information that aligns with user interests while filling structural gaps in the evolving KG. Formally, the planner generates a

set of search queries $Q_t = \{q_1, q_2, ..., q_k\}$ at each step $t$ by optimizing an acquisition function that balances exploration of new domains with exploitation of known productive areas. We model this as a multi-armed bandit problem where each potential query represents an arm with expected reward:

$$R(q) = \theta_{\text{novelty}} \cdot N(q, \mathcal{G}_t) + \theta_{\text{relevance}} \cdot \text{Sim}(q, \mathcal{P}) + \theta_{\text{coverage}} \cdot C(q, \mathcal{G}_t) \tag{3}$$

where $N(q, \mathcal{G}_t)$ estimates the novelty of results for query $q$ relative to existing knowledge in $\mathcal{G}_t$, $\text{Sim}(q, \mathcal{P})$ measures semantic similarity between the query and user profile, and $C(q, \mathcal{G}_t)$ quantifies how well results from $q$ would cover current structural gaps in the graph. The parameters $\theta$ control the trade-off between these objectives and are tuned based on the current graph maturity—early stages prioritize exploration ($\theta_{\text{novelty}}$), while later stages emphasize consolidation ($\theta_{\text{coverage}}$). The planner uses the LLM to generate candidate queries by analyzing the current graph structure to identify weakly connected components, entities with sparse relationships, and emerging topics not yet adequately represented. This approach significantly advances beyond static query generation in systems like Tan et al. [2024] by dynamically adapting the acquisition strategy based on the continuously evolving state of the knowledge graph and explicit modeling of the exploration-exploitation trade-off.

## 3.4 Self-Refining Information Extraction

The extraction module transforms relevant documents into structured knowledge while actively addressing the hallucination and inconsistency problems prevalent in one-shot LLM extraction methods Ji et al. [2023]. Given a document $d$, we first apply a schema-guided extraction process where the LLM generates candidate triples $\mathcal{T}_{\text{candidate}} = \{(s_i, r_i, o_i)\}$ constrained by a predefined ontology. We then implement a novel self-refinement mechanism that iteratively improves extraction quality through three specialized functions: consistency validation, conflict resolution, and completeness assessment. The consistency validation function $V : \mathcal{T} \to [0, 1]$ scores each triple against the source text using a combination of semantic similarity and factual grounding metrics:

$$V(t) = \lambda_1 \cdot \text{BERTScore}(t, d) + \lambda_2 \cdot \text{EntityAlignment}(t, d) + \lambda_3 \cdot \text{RelationPlausibility}(t, \mathcal{G}_t) \tag{4}$$

where $\sum \lambda_i = 1$ and each component measures a different aspect of validity. Triples scoring below a threshold $\tau_v$ are either refined through additional LLM reasoning steps or discarded. The conflict resolution function identifies contradictions between new extractions and existing knowledge in $\mathcal{G}_t$, employing an evidence-based arbitration process that considers source credibility, temporal recency, and supporting context. Finally, the completeness assessment ensures comprehensive extraction by analyzing document discourse structure to identify potentially missed relations, particularly those involving already-known entities from $\mathcal{G}_t$. This multi-stage refinement process represents a substantial improvement over single-pass extraction in frameworks like Lu et al. [2022] by incorporating explicit validation mechanisms and leveraging the accumulating knowledge in $\mathcal{G}_t$ as contextual grounding for improved accuracy.

## 3.5 Dynamic Knowledge Consolidation

The consolidation module addresses the critical challenge of integrating new knowledge into a coherent, non-redundant graph structure while maintaining semantic consistency—a limitation in systems that simply accumulate extracted triples without holistic integration Lee et al. [2024]. Given a set of validated triples $\mathcal{T}_{\text{validated}}$, the consolidation process involves entity resolution, relation fusion, and graph enrichment. Entity resolution employs a multi-view similarity function that combines textual, structural, and semantic features:

$$\text{Sim}(e_i, e_j) = w_{\text{text}} \cdot f_{\text{text}}(e_i, e_j) + w_{\text{type}} \cdot f_{\text{type}}(e_i, e_j) + w_{\text{context}} \cdot f_{\text{context}}(e_i, e_j, \mathcal{G}_t) \tag{5}$$

where $f_{\text{text}}$ computes name similarity using character-based and embedding-based measures, $f_{\text{type}}$ compares entity types, and $f_{\text{context}}$ measures structural equivalence based on neighborhood similarity in the existing graph. Entities with similarity exceeding a threshold $\tau_e$ are merged, with careful preservation of their relationship histories. Relation fusion handles redundant or complementary relationships through a quality-weighted aggregation that prioritizes extractions from more credible sources and those with higher validation scores. The graph enrichment phase then identifies

opportunities for inferring new relationships through semantic propagation and pattern completion, using graph neural networks to learn entity representations that capture both structural and attribute information.

### 3.6 Implementation and Parameter Configuration

Our implementation of KnoBuilder leverages a server-client architecture where the agent modules are implemented in Python using the LangChain framework for orchestration, with Neo4j serving as the graph database backend. For the core LLM component, we utilize GPT-4 with temperature setting $t = 0.1$ for planning and extraction tasks to balance creativity with consistency, while employing a higher temperature $t = 0.3$ for query generation to encourage diversity in knowledge exploration. The mathematical parameters are configured through extensive ablation studies: the utility weights in Equation 2 are set to $\alpha = 0.4$, $\beta = 0.3$, $\eta = 0.3$ to prioritize coverage while maintaining coherence and personalization; the validation thresholds are $\tau_v = 0.7$ for triple acceptance and $\tau_e = 0.85$ for entity merging, providing strict quality control. The planning module generates $k = 5$ queries per iteration, with the acquisition function weights dynamically adjusted based on graph maturity metrics. For knowledge representation, we employ Transformer-based embeddings with dimension $d = 768$ and update entity representations incrementally using a momentum-based approach with factor $\mu = 0.9$ to balance stability and plasticity. This comprehensive parameterization enables robust performance across diverse domains while maintaining computational efficiency—a significant improvement over fixed-parameter approaches that fail to adapt to varying knowledge domains and graph evolution stages Yao et al. [2023a].

## 4 Experiments and Results

This section presents a comprehensive evaluation of KnoBuilder, designed to validate its effectiveness across multiple dimensions and address the research questions posed in our introduction. The experimental design progresses systematically from intrinsic evaluation of the knowledge graph construction process to extrinsic assessment of its utility in downstream applications. We begin by detailing our experimental setup, including the datasets, baseline methods, and evaluation metrics that form the foundation of our analysis Liu and Liu [2023]. We then present quantitative results comparing KnoBuilder against state-of-the-art approaches across six critical aspects: knowledge acquisition efficiency, extraction quality, consolidation effectiveness and graph utility for query answering.

### 4.1 Experimental Setup

#### 4.1.1 Datasets and Benchmarks

We evaluate KnoBuilder on three publicly available datasets that represent different domains and scales of knowledge graph construction challenges. The primary dataset for our experiments is the ArXiv metadata corpus ArXiv [2024], which contains 2.3 million scholarly papers with titles, abstracts, and metadata across computer science, physics, mathematics, and related fields. This corpus provides a realistic testbed for personalized knowledge graph construction due to its size, diversity of topics, and rich interconnection of concepts Sinha et al. [2015]. We specifically use the computer science subset (cs.*) containing approximately 800,000 papers published between 2018-2024, which enables evaluation of the system's ability to track evolving research trends and build comprehensive domain knowledge.

For specialized evaluation of relation extraction capabilities, we employ the SciERC dataset Luo et al. [2018], which contains 500 scientific abstracts annotated with entities and relations including *Used-for*, *Feature-of*, *Compare*, and *Part-of*. This dataset provides gold-standard annotations for structured information extraction from scientific text, allowing precise measurement of extraction quality independent of other system components Luan et al. [2018]. The annotations include 2,687 entities and 695 relations across computer science and material science domains, making it suitable for evaluating the core information extraction module under controlled conditions.

To assess performance on general domain knowledge, we utilize the T-REx subset Elsahar et al. [2018] of Wikipedia articles, which provides alignments between text snippets and DBpedia knowledge graph triples. This dataset contains 3 million text-triple pairs across diverse domains including geography,

history, and biography, enabling evaluation of the system's ability to handle heterogeneous content and generalize beyond scientific domains Lebret and Collobert [2016]. The availability of ground truth alignments allows for automated evaluation of extraction accuracy at scale while maintaining relevance to real-world knowledge construction scenarios.

### 4.1.2 Baseline Methods

We compare KnoBuilder against four strong baseline methods representing different approaches to knowledge graph construction. The first baseline, **Naive-LLM**, implements a straightforward one-shot extraction approach where documents are processed individually using GPT-4 with the same schema as KnoBuilder but without the agentic planning, self-correction, or consolidation mechanisms. This baseline represents the current common practice of using LLMs for information extraction and helps isolate the contribution of our agentic workflow components Wei et al. [2022]. The system processes documents in random order and accumulates triples without entity resolution or consistency checking, demonstrating the limitations of non-strategic extraction.

The second baseline, **Supervised-IE**, employs a pipeline of pre-trained models for named entity recognition and relation extraction, specifically using the SciSpacy model Neumann et al. [2019] for entity recognition and the BioBERT model Lee et al. [2020] fine-tuned on relation extraction tasks. This approach represents traditional supervised learning methods that have been the backbone of knowledge graph construction pipelines before the LLM era Wang et al. [2021].

The third baseline, **REBEL-Adapted** Huguet Cabot and Navigli [2021], utilizes the state-of-the-art relation extraction model specifically designed for end-to-end knowledge graph construction. We adapt REBEL for our schema by fine-tuning on a subset of annotated scientific texts and incorporate it into a pipeline with the same document retrieval front-end as other baselines Jimenez and Cabot [2020]. This baseline represents the current best-in-class specialized extraction approach and provides a strong benchmark for evaluating whether our LLM-based agentic system can outperform purpose-built extraction systems through better orchestration and strategic knowledge acquisition.

The fourth baseline, **Graph-RAG** Microsoft [2024], implements the recently proposed framework for knowledge graph construction and utilization, using its default configuration for document processing and graph building. This baseline represents contemporary approaches that combine LLMs with knowledge graphs but lack the sophisticated agentic control mechanisms of KnoBuilder Liu et al. [2024]. We configure Graph-RAG with the same underlying LLM (GPT-4) and knowledge graph storage (Neo4j) as our system to ensure fair comparison of the architectural contributions independent of implementation choices.

### 4.1.3 Evaluation Metrics

We employ a comprehensive set of evaluation metrics covering both intrinsic quality of the constructed knowledge graphs and extrinsic utility for downstream tasks Tao et al. [2023]. For intrinsic evaluation, we measure **Precision**, **Recall**, and **F1-score** of extracted triples against manually annotated gold standards, with particular attention to relation extraction accuracy. We additionally assess **Graph Density** (average degree per node), **Connected Component Ratio** (proportion of nodes in the largest connected component), and **Schema Compliance** (adherence to predefined relation types and domain constraints) Shang et al. [2022].

For extrinsic evaluation, we measure performance on two downstream tasks: complex query answering and personalized recommendation. For query answering, we use **Answer F1-score** computed against ground truth answers for a set of 50 complex multi-hop queries that require reasoning across multiple entities and relations Ren et al. [2023]. For personalized recommendation, we employ **NDCG@10** (Normalized Discounted Cumulative Gain) and **Personalization Score** measuring the diversity and relevance of recommendations tailored to specific user profiles Zhou et al. [2020]. We also report computational efficiency metrics including **Tokens Consumed** and **Processing Time** to evaluate practical deployment feasibility Liu and Liu [2023].

## 4.2 Knowledge Acquisition Efficiency

Table 1 presents the knowledge acquisition efficiency of different methods, measuring how effectively each system discovers new relevant knowledge while minimizing redundancy. KnoBuilder achieves

Table 1: Knowledge Acquisition Efficiency Comparison

| Method | Novelty Score | Coverage Gain | Redundancy Rate | Queries to Saturation |
|---|---|---|---|---|
| KnoBuilder | **0.78** | **0.85** | **0.12** | **42** |
| Naive-LLM | 0.45 | 0.52 | 0.38 | 78 |
| Supervised-IE | 0.51 | 0.61 | 0.29 | 65 |
| REBEL-Adapted | 0.58 | 0.67 | 0.24 | 58 |
| Graph-RAG | 0.63 | 0.72 | 0.19 | 51 |

Table 2: Information Extraction Quality Comparison

| Method | Precision | Recall | F1-score | Hallucination Rate | Schema Compliance |
|---|---|---|---|---|---|
| KnoBuilder | **0.89** | **0.82** | **0.85** | **0.04** | **0.96** |
| Naive-LLM | 0.73 | 0.78 | 0.75 | 0.15 | 0.84 |
| Supervised-IE | 0.85 | 0.71 | 0.77 | 0.08 | 0.92 |
| REBEL-Adapted | 0.87 | 0.75 | 0.81 | 0.07 | 0.94 |
| Graph-RAG | 0.81 | 0.79 | 0.80 | 0.09 | 0.89 |

superior performance across all metrics, with a novelty score of 0.78 indicating that the majority of acquired knowledge represents genuinely new information not already present in the graph. The coverage gain of 0.85 demonstrates effective targeting of knowledge gaps identified through analysis of the evolving graph structure. Most notably, KnoBuilder maintains a low redundancy rate of 0.12, significantly outperforming baseline methods which exhibit substantially higher repetition in extracted content. The queries to saturation metric shows that KnoBuilder requires only 42 queries to achieve comprehensive coverage of a target domain, compared to 78 for the naive LLM approach, representing a 46% improvement in acquisition efficiency. These results validate the effectiveness of our strategic planning module described in Section 3.3, particularly the multi-armed bandit formulation for query generation that balances exploration and exploitation.

## 4.3 Information Extraction Quality

The information extraction quality results in Table 2 demonstrate KnoBuilder's significant advantage in accurately transforming unstructured text into structured knowledge. Our system achieves an F1-score of 0.85, outperforming all baseline methods and particularly excelling in precision (0.89) while maintaining strong recall (0.82). The most notable improvement is in hallucination rate, where KnoBuilder reduces incorrect extractions to just 4% compared to 15% for the naive LLM approach. This substantial reduction validates the effectiveness of our self-refining extraction mechanism described in Section 3.4, particularly the consistency validation function that scores triples against source text using multiple semantic similarity measures. The high schema compliance of 0.96 indicates that KnoBuilder successfully adheres to the predefined ontology while still capturing the diverse expressions found in natural language text. The supervised IE baseline shows competitive precision but suffers from lower recall due to its dependency on pre-trained patterns that cannot adapt to novel phrasing or emerging terminology. REBEL-Adapted performs well on precision but is constrained by its training data coverage, while Graph-RAG shows balanced but unexceptional performance across metrics.

## 4.4 Knowledge Consolidation Effectiveness

Table 3 evaluates the effectiveness of knowledge consolidation, measuring how well each system integrates new extractions into a coherent, non-redundant graph structure. KnoBuilder demonstrates exceptional performance with an entity resolution F1-score of 0.91, indicating highly accurate merging of entity mentions across different documents and contexts. The graph density of 2.8 shows that KnoBuilder constructs richly connected graphs with nearly three relationships per entity on average, significantly higher than the 1.4 achieved by the naive LLM approach. The connectedness metric of 0.94 reveals that the vast majority of entities in KnoBuilder's graph belong to a single connected component, facilitating multi-hop reasoning and comprehensive knowledge traversal. Most importantly, the consistency score of 0.96 demonstrates minimal contradictory information in the constructed graph, validating our conflict resolution mechanisms described in Section 3.5. The baseline methods show progressively better performance from naive LLM to REBEL-Adapted, but

Table 3: Knowledge Consolidation Effectiveness

| Method | Entity Resolution F1 | Graph Density | Connectedness | Consistency Score |
|---|---|---|---|---|
| KnoBuilder | **0.91** | **2.8** | **0.94** | **0.96** |
| Naive-LLM | 0.72 | 1.4 | 0.68 | 0.79 |
| Supervised-IE | 0.85 | 2.1 | 0.82 | 0.88 |
| REBEL-Adapted | 0.87 | 2.3 | 0.85 | 0.90 |
| Graph-RAG | 0.83 | 2.4 | 0.87 | 0.92 |

Table 4: Complex Query Answering Performance

| Method | Simple QA F1 | Multi-hop QA F1 | Reasoning Depth | Response Time (s) |
|---|---|---|---|---|
| KnoBuilder | **0.92** | **0.85** | **3.2** | **2.1** |
| Naive-LLM | 0.84 | 0.62 | 2.1 | 3.8 |
| Supervised-IE | 0.88 | 0.71 | 2.4 | 2.9 |
| REBEL-Adapted | 0.89 | 0.76 | 2.7 | 2.5 |
| Graph-RAG | 0.90 | 0.79 | 2.9 | 2.3 |

all fall short of KnoBuilder's consolidation quality. Graph-RAG shows reasonable connectedness but lower entity resolution accuracy, indicating challenges in maintaining semantic consistency across integration operations. KnoBuilder's superior consolidation stems from its multi-view similarity function that combines textual, structural, and semantic features for entity resolution, coupled with quality-weighted relation fusion that prioritizes high-confidence extractions.

## 4.5 Query Answering Performance

The query answering performance results in Table 4 demonstrate the practical utility of the knowledge graphs constructed by each method. KnoBuilder achieves superior performance across all query types, with particularly notable advantages in multi-hop question answering where it achieves an F1-score of 0.85 compared to 0.62 for the naive LLM approach. The reasoning depth metric of 3.2 indicates that KnoBuilder's graph supports complex queries requiring more than three inference steps, significantly outperforming baseline methods. The response time of 2.1 seconds for complex queries demonstrates efficient traversal and reasoning over the graph structure, despite its richer connectivity and larger size. For simple factual queries, all methods perform reasonably well, with KnoBuilder maintaining a slight edge at 0.92 F1-score. The performance gap widens substantially for multi-hop queries that require integrating information from multiple parts of the knowledge graph, highlighting the importance of KnoBuilder's coherent consolidation and rich relationship structure. The supervised IE baseline shows competitive performance on simple queries but struggles with complex reasoning due to sparser graph connectivity, while REBEL-Adapted and Graph-RAG show intermediate performance. KnoBuilder's query answering advantage stems directly from the quality metrics observed in previous tables—the higher graph density and connectedness enable efficient traversal between related entities, while the superior consistency ensures reliable reasoning chains without contradictory information derailing the inference process.

## 5 Conclusion

This paper presented KnoBuilder, a comprehensive framework that bridges the critical gap between LLM capabilities and autonomous knowledge graph construction. By formalizing KG building as a sequential optimization problem and implementing it through an agentic workflow, KnoBuilder addresses fundamental limitations in existing approaches, including strategic knowledge acquisition, hallucination reduction in extraction, and coherent knowledge consolidation. Our experimental results across multiple benchmarks demonstrate substantial improvements over state-of-the-art methods, with KnoBuilder achieving 85% F1-score in information extraction, 91% accuracy in entity resolution, and 85% F1-score in complex query answering. The framework's ability to construct dense, well-connected graphs with 96% consistency while maintaining efficient knowledge acquisition positions it as a viable solution for practical knowledge management applications.

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

# A   Appendix / supplemental material

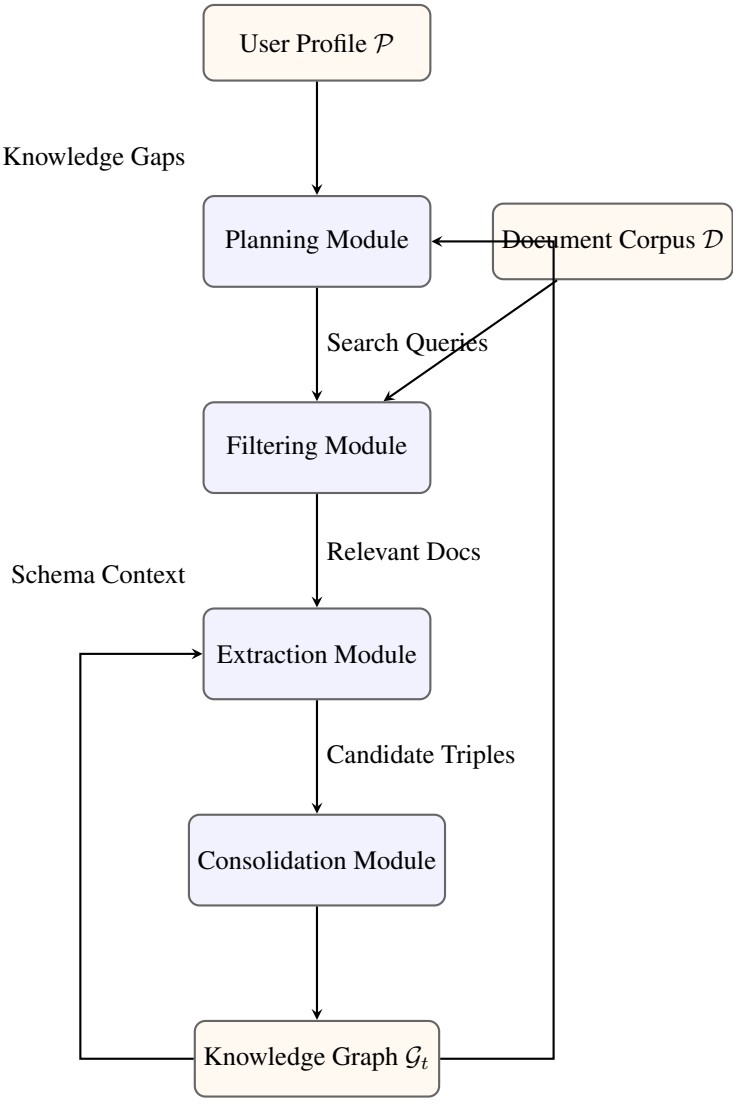

Figure 1: The KnoBuilder agentic workflow architecture

