# OpenReview forum: "[Regular] KnoBuilder: An LLM-Agent for Autonomous and Personalized Knowledge Graph Construction from Unstructured Text"
_NeurIPS.cc/2025/Workshop_Mexico_City/NORA — NeurIPS 2025 Workshop NORA Poster_

### Official Review · Reviewer_Rqzb · 2025-11-01
**Good for a proof-of-concept, though more details are appreciated.**

**Rating:** 5
**Confidence:** 4

**Review:**

This paper proposes KnoBuilder, a framework for creating, curating, and maintaining a knowledge graph based on a theme or user profile. It shows that the proposed pipeline is reasonably accurate and practical, as expected. The paper also goes out of its way to mathematically formalize the framework as an optimization problem. Although it does not seem to be the focus of the paper, it is nonetheless a welcome addition.

Novelty: the idea itself is not new, as a few commercial companies (Resolve.ai) have implemented and provide services for agentic AI workflow in knowledge graph curation. However, in terms of academic discourse, having a document to be referenced is good, as an academic groundwork.

Clarity: The paper is mostly well-written, with the training details (e.g., the knowledge representation using transformer-based embedding) are provided, and the sections are easy to read. That said, there are a few places where terms used are unclear. For example, the Reasoning Depth metric in Table 4 is not mentioned in the list of evaluation metrics in Section 4.1.3. It is also not explained how it is calculated for each of the model. There also doesn't seem to be any error analysis of the output, although given the main contribution being an academic reference of an agentic AI applied on KG curation, that seems passable.

Reproducibility: one big weakness of this paper is the fact that the prompts used in the LLM (as part of the agentic AI flow) are not shared. It's impossible to replicate nor reproduce the result without the prompts, as the quality of the prompts heavily affects the performance of the system, not just for the proposed KnoBuilder, but also all other baselines. Only the dataset (ArXiv metadata corpus) and theme (cs.*) is mentioned.

Ethical Compliance: There doesn't seem to be any ethical concerns with this work, beyond general LLM usage, as most probably used by any other paper here.

---

### Official Review · Reviewer_qDEt · 2025-11-06
**Interesting Agentic Framework, but Too Abstract and Under-Specified for a Scientific Pape**

**Rating:** 3
**Confidence:** 4

**Review:**

The paper presents an interesting agentic framework for personalized KG construction and a useful high-level problem formulation. However, the manuscript lacks critical implementation and evaluation details necessary for reproducibility: precise formulas for the utility components, the learning/update algorithm for the policy, exact prompt texts and LLM call details, and the datasets/protocols used for downstream evaluation (particularly personalization and multi-hop QA).

Several major issues require substantial revision:

1. **High-level descriptions only; key formulas unspecified**
   The three utility components (Coverage / Coherence / Personalization) are presented conceptually, but the paper does not provide precise, operational formulas or normalization conventions. These quantities directly determine rewards and therefore change the behavior and evaluation outcomes; omitting exact definitions prevents reproducibility.

2. **Policy learning / update procedure not specified**
   The manuscript states a “POMDP-style” formulation and mentions “bandit-style” acquisition, but it does not specify which algorithm was used in practice, how the policy is parameterized, how often updates occur, what hyperparameters (learning rates, batch sizes) were used, or how credit assignment was performed for multi-query actions.

3. **Prompts and LLM interface details missing**
   GPT-4 is reported as the core component, yet the paper omits exact prompt texts (system/user/assistant messages), context window usage, chunking strategy, temperature choices associated with specific prompts, and any prompt engineering that materially affects extraction quality. Given how crucial prompts are for LLM behavior, this is a serious gap for anyone attempting to reproduce the system.

4. **Evaluation protocol and data missing for downstream tasks**
   The paper lists downstream tasks such as “personalized recommendation” and “50 multi-hop queries,” but it fails to specify the datasets or gold standards used for these evaluations (what user profiles were used, how relevance labels were defined for recommendation, what queries exactly were used for multi-hop QA). Results without dataset/protocol details are not verifiable.

In addition, there are issues with the references that further reduce clarity and credibility. For example (this list is not necessarily exhaustive):

- The survey “Li [2025]” is cited in the literature review as articulating a broader vision of LLMs for data management and KG construction, but the corresponding reference is *Zichao Li, “MCL for MLLMs: Benchmarking forgetting in task-incremental multimodal learning,” ICCV 2025*, which is about multimodal continual learning rather than data management or knowledge graphs.

- “Liu and Liu [2023]” is cited as a general reference for the experimental design and for computational efficiency metrics in this paper’s evaluation setup, but the reference entry is *“Evaluating large language models trained on code” (arXiv:2307.03374)*, which is not about knowledge graph evaluation or system efficiency and seems only tangentially related.

Overall, while the high-level idea is interesting, the current manuscript lacks sufficient detail and consistency for readers to reimplement the method or reproduce the experiments. A major revision is required before the work can be considered ready for publication.

---

### Official Review · Reviewer_Zf75 · 2025-11-06
**Promising Framework with Room for Technical Clarity and Reproducibility Enhancement**

**Rating:** 4
**Confidence:** 3

**Review:**

This paper introduces an LLM-based agentic framework designed for the autonomous and personalized construction of knowledge graphs (KGs) from unstructured text, addressing the limitations of traditional KG methods and one-shot LLM extraction approaches that suffer from inconsistency and lack of strategic focus. The framework operates through a synergistic loop between an LLM agent and a dynamically evolving KG, featuring strategic planning for knowledge acquisition, self-refining information extraction with multi-stage validation, and dynamic consolidation to maintain graph coherence. Evaluations on scientific corpora demonstrate KnoBuilder's superior performance, achieving an 85% F1-score in extraction quality, a 46% improvement in acquisition efficiency, 91% entity resolution accuracy, and enhanced complex query answering, while maintaining coherent graph structures with 96% consistency scores.

The authors are tackling the important research challenge in autonomous knowledge graph construction using LLM-based agents. The proposed method is considered useful as it demonstrates superior performance to existing methods in experimental results. However, to strengthen the paper for acceptance, I would appreciate your consideration of the following points for improvement.
- Highlighting the specific differences between each module and existing techniques would better emphasize the originality of this research. In particular, more detailed explanations of the algorithmic innovations in the self-refining mechanism and dynamic consolidation would be valuable.
- Consider providing implementation details, particularly the specific prompt designs for the LLM and error handling methods, in supplementary materials.
- Including the results of the "extensive ablation studies" mentioned in Section 3.6 would clarify the validity of parameter choices and deepen understanding of the method.

---

### Official Review · Reviewer_w9XQ · 2025-11-06
**Consolidated multi-step knoweldge graph construction from unstructured text**

**Rating:** 7
**Confidence:** 4

**Review:**

This paper presents an architecture consisting of a pipeline of modules for building and enriching a knowledge graph in an iterative way by taking into account the user profile. The modules are the following:

1. Planning: generating queries based on user profile and state of graph;

2. Filtering: assessing relevance of retrieved docs according to a multifaceted scoring function that takes into account novelty, source credibility and topic alignment;

3. Extraction: extracting triples, ensuring consistency, resolving resolution and assessing completeness, where consistency considers similarity of triple against source text, entity alignment and relation plausibility;

4. Consolidation: integrating the extracted triples in the knowledge graph, resolving entity conflict and reinforcing connection through semantic similarity measures.

The workflow iterates until assigned resources budget is exhausted or no further improvements to the KG is measured.

The architecture is tested with 2 datasets: the arXiv cs subset (together with a dataset of 500 abstracts annotated with entities and relations) and T-rex (which provides alignments between text snippets and DBPedia KG triples) and also tested extrinsically with a question answering task. KnoBuilder is evaluated against a naive LLM baseline, Supervised IE, and state of the art REBEL-Adapted and Microsoft GraphRAG, which it outperforms in all criteria considered, and also downstream for question answering.

The paper is well written and readable and every module implements several measures to ensure it achieves the best result. The approach is tested against different approaches including state of the art.

The authors could have performed some ablation studies to show the benefit of the different measures and approaches used.  Also there is no self critic of the work, like addressing the limitations and how could this work be improved.

Some of the references seem to be preprints without reviews which I think should be avoided in a conference.

Some of the text is a bit 'lax' for a scientific conference paper: "an autonomous AI" (line 34), "LLMs with their extensive world knowledge" (lines 55-56),

Some terms used are a bit unclear, like the fact that the extraction module is self-correcting or self-refining (both terms are used).  Even the term 'agent' is unclear in my opinion as I see the approach like an iterative pipeline with modules, but I guess this is getting into the polemic of what AI agents mean and how broad the concept is.

Some details could be added like: what is the user profile as input, what is the schema? How big is the KG at different iterations? (number of triples and relations). Maybe even showing an example KG with initial triples and addons could help.

Small detail: Figure 1 is very helpful but could be improved (the arrows going from Document Corpus are all mingled together).

---

### Official Review · Reviewer_9LLN · 2025-11-06
**Comprehensive approach to sequential self-correcting optimization of knowledge graphs**

**Rating:** 7
**Confidence:** 3

**Review:**

The paper proposes an architecture workflow for creating KG based on unstructured data and compared to similar recent work on agentic KG building, the authors include the “entire pipeline” of personalized KG creation in one agentic loop. The methodology is explained in detail for each module, and results based on a number of metrics compared to multiple baselines, show considerable improvements.
Overall I enjoyed reading the paper and the proposed approach seems to be effective and practical as well. I also appreciated that the authors included comparison of each separate module in the pipeline to related work, which makes the contributions clearer.
I just recommend adding future directions in the conclusions.

---

### Official Review · Reviewer_EAoi · 2025-11-07
**Very interesting idea but completely devoid of actual details**

**Rating:** 2
**Confidence:** 5

**Review:**

I find the idea compelling but the paper is lacking a discussion of the methodology (why, how did you decide to pick the specific mathematical formulation, what else was tried, etc) but most importantly the lack of indication that the code is published as part of the publication and the lack of accurate description of implementation choices for each component, as well as key details (e.g., "Transformer-based embeddings"), make the claims made by the authors non-reproducible and a matter of faith.

I think if the actual methods were explained carefully providing key details, and the choices made argumented more clearly this could be a valuable paper.

---

### Official Review · Reviewer_J7k1 · 2025-11-07
**Paper introduces a novel agentic framework for personalized Knowledge graph construction from unstructured document corpora. The framework implements a synergistic loop between  an LLM agent and an evolving knowledge graph, guided by a user profile, with key steps for planning, filtering, extraction, self-correction and consolidation. Experimental evaluation on scientific corpora demonstrate the benefits of the approach through a variety of metrics.**

**Rating:** 8
**Confidence:** 4

**Review:**

Paper presents the KnoBuilder framework that constructs personalized knowledge graph from documents through synergistic combination of an LLM agent and evolving Knowledge Graph, and is very relevant to the NORA workshop audience.

Key contributions include
S1. A mathematical framework for the agentic workflow consisting of an iterative, state maintained flow consisting of four modules:
(i) planning module to identify strategic gaps and exploratory directions based on user profile and current knowledge graph state.
(ii) filtering module to process documents of interest using a scoring function that considers topical alignment, novelty relative to existing knowledge and source credibility.
(iii) extraction module to extract structured triples with emphasis on minimizing hallucinations and inconsistencies.
(iv) consolidation module to integrate new knowledge into existing graph while maintaining semantic consistency. This step includes entity resolution, relation fusion and graph enrichment steps.

S2: Experimental results are presented over three publicly available datasets comparing against four baselines covering techniques including one-shot LLM based extraction, supervised pre-trained information extraction models, specialized relation extraction models and a recently proposed framework for knowledge graph construction leveraging LLMs. Intrinsic and extrinsic evaluation metrics are used to demonstrate the benefits of the proposed framework. Highlights include
(i) Knowledge acquisition efficiency is higher for KnoBuilder with better novelty and coverage, lower redundancy score and fewer queries to saturation.
(ii) Quality of extracted information is better with higher F1 scores, lower hallucination rates and higher schema compliance.
(iii) Knowledge consolidation effectiveness is better with higher entity resolution F1, higher graph density, better connectedness and higher consistency score.
(iv) Query answering performance is better for simple QA F1, with increased benefits for multi-hop QA F1, lower response time and higher reasoning depth.


Opportunities to improve the paper include
O1: Distinction between the planner and filtering modules is unclear. Section 3.3 describes the planning module as incorporating novelty, relevance and coverage, and this output is directly processed by a filtering module, which seems to be plain retrieval of the documents from the underlying corpora. Section 3.2 described the filtering module as a separate step that considers topical alignment, novelty relative to existing knowledge and source credibility. The corresponding details are missing in the paper.

O2: The extraction module refers to a predefined ontology. How is this ontology constructed and maintained ?

O3: How are the Supervised-IE and REBEL-Adapted baselines used for knowledge graph construction; details of how the extracted tuples are consolidated into a knowledge graph is missing. Consequently, references to hallucination rate and schema compliance in Table 2 for these baselines are not clear.

O4: Experimental section is hard to follow - e.g., while three datasets are mentioned in Section 4.1.1, how they correspond to the results described in subsequent sections and Tables 1-4 is not clear. Some of the references in Section 4.1.3, such as extrinsic evaluation for personalized recommendation and computational efficiency metrics, are not elaborated in the rest of the paper.

O5: Examples that illustrate where the proposed techniques make the knowledge graph more consistent and relevant based on the "user profile" would make it easier for the reader to follow the key contributions better.